# Cognitive-behavioral therapy for the improvement of negative symptoms and functioning in schizophrenia: A systematic review and meta-analysis of randomized controlled trials

**Yu Hong**[1]*, **Yiyun Chen**[2], **Yinglin Bai**[1], **Wenfei Tan**[1]

**1** School of Health and Nursing, Guangzhou Huali College, Guangzhou, China, **2** Department of Law, Shantou University, Shantou, China.

* 13026890362@163.com

## Abstract

### Background

Negative symptoms of schizophrenia are a range of deficits or losses in mental functioning associated with the disorder, including blunted affect, alogia, avolition, asociality, and anhedonia. These symptoms severely impact the quality of life of patients and hinder the recovery process. They significantly impair patients' ability to live independently, maintain social relationships, and function effectively in society. However, current treatments for negative symptoms of schizophrenia are limited in efficacy and remain controversial. Cognitive-behavioral therapy (CBT) is a goal-oriented psychotherapy that aims to improve individuals' emotional and psychological states by changing their negative thought patterns and behaviors. It helps patients identify and challenge irrational beliefs while promoting more positive behavioral changes through behavioral experiments and skills training. This study aims to conduct a meta-analysis to assess the effects of CBT on negative symptoms and function in schizophrenia.

### Objectives

This study aimed to investigate the effects of cognitive behavioral therapy on negative symptoms, mental function, social skills, and social functioning in schizophrenia.

### Methods

Literature was retrieved from 10 databases (PubMed, EMBASE, Cochrane Library, Web of Science, APA PsycINFO, CINAHL, MEDLINE, CNKI, Wan fang Database and SinoMed,), with the search period ranging from the inception date to 1 September 2024. Two researchers independently conducted a literature review, data extraction, and risk of bias assessment. The quality of the included studies was assessed using

**Data availability statement:** All relevant data are within the paper and its Supporting Information files.

**Funding:** The author(s) received no specific funding for this work.

**Competing interests:** The authors have declared that no competing interests exist.

the Cochrane Risk of Bias tool, and the meta-analysis was conducted using RevMan 5.3. The measurement outcomes include negative symptoms of schizophrenia, overall function, social skills, and social functioning.

## Result

The analysis included a total of 15 studies involving 1,311 participants. All studies used the Positive and Negative Syndrome Scale (PANSS) as the assessment tool for measuring negative symptoms of schizophrenia. The results of the meta-analysis indicated that cognitive-behavioral therapy (CBT) significantly improved negative symptoms in patients with schizophrenia compared to treatment as usual (TAU) (MD = -1.65, 95% CI = -2.10 to -1.21, $p < 0.001$, $I^2 = 41\%$). Short-term CBT significantly improved negative symptoms in schizophrenia (MD = -2.71, 95% CI = -3.18 to -1.61, $p < 0.001$, $I^2 = 48\%$). Medium-term CBT also significantly improved negative symptoms (MD = -1.80, 95% CI = -2.76 to -0.84, $p < 0.001$, $I^2 = 29\%$). Long-term CBT demonstrated significant improvement in negative symptoms as well (MD = -1.70, 95% CI = -2.54 to -0.85, $p < 0.001$, $I^2 = 0\%$). CBT significantly improved overall function in patients with schizophrenia (SMD = 0.38, 95% CI = 0.13 to 0.63, $p < 0.05$, $I^2 = 0\%$). Additionally, CBT significantly enhanced social skills (SMD = 0.87, 95% CI = 0.58 to 1.16, $p < 0.001$, $I^2 = 0\%$) and social functioning (SMD = 0.19, 95% CI = 0.03 to 0.36, $p < 0.05$, $I^2 = 24\%$) in these patients.

## Conclusion

The results indicate that cognitive behavioral therapy has a significant effect on improving the negative symptoms of schizophrenia and is markedly superior to Treatment as Usual (TAU). Moreover, all three sub-treatment approaches (short-term, medium-term, and long-term) can sustainably and significantly improve negative symptoms of schizophrenia. Future research should focus on developing and evaluating cognitive therapies targeting negative symptoms, providing more reliable evidence and applying these research findings to clinical practice.

## Introduction

Schizophrenia is a profound and enduring mental health condition that profoundly impacts an individual's cognition, perception, emotional responses, and actions [1]. The reach of its effects is extensive, not only deeply affecting the lives of those who live with the disorder but also significantly impacting their families and support networks [2].

Negative symptoms are an important aspect of schizophrenia, contrasting with positive symptoms such as hallucinations and delusions. They include blunted affect, alogia, avolition, asociality, and anhedonia [3,4], which can significantly impair patients' quality of life and hinder the rehabilitation process [5,6].When treating schizophrenia, in addition to focusing on controlling positive symptoms, appropriate management and intervention for negative symptoms are also necessary.

The condition can give rise to a range of challenges, including disruptions in social interactions, hindered professional capabilities, and a diminished overall quality of life. Schizophrenia interventions encompass a comprehensive array of strategies designed to mitigate symptoms, bolster functional capabilities, and elevate the overall quality of life for those affected. Antipsychotic medications are the primary intervention for the treatment of schizophrenia. However, current evidence indicates that the efficacy of antipsychotic medications in reducing the severity of negative symptoms is highly limited, and there is a lack of robust evidence for their effectiveness in treating primary and enduring negative symptoms [7–9]. This represents a core challenge in current research on the treatment of negative symptoms, while also highlighting the insufficient evidence supporting existing interventions. In the clinical treatment of schizophrenia, we are gradually exploring ways to reduce reliance on traditional antipsychotic medications [10], instead focusing on measures of cognitive behavioral therapy to improve the condition.

In the field of schizophrenia treatment, Cognitive Behavioral Therapy (CBT) has demonstrated multidimensional intervention characteristics and has been widely applied. As a structured and goal-oriented psychotherapy, CBT has garnered significant attention in the treatment of schizophrenia in recent years [11–13], aimed at improving patients' cognitive functions, reducing symptoms, enhancing the quality of life, and promoting the recovery of social functioning. Previous review studies have often focused on the improved effects of cognitive behavioral therapy on the positive symptoms of schizophrenia [14,15]. However, as recent studies have pointed out [16], there is still a lack of comprehensive analyses specifically addressing the improvement of negative symptoms. Xu's study [16] has provided important insights into the efficacy of CBT in alleviating negative symptoms. Building on this foundation, our study further expands the research in this area. Specifically, we have increased the number of included studies, clearly distinguished the short-, medium-, and long-term effects of CBT, limited the control group to treatment as usual (TAU), and introduced function as an important outcome measure. To gain a deeper understanding of the efficacy of cognitive-behavioral therapy (CBT) in treating negative symptoms of schizophrenia, systematic reviews and meta-analyses are essential. They not only fill the existing research gaps but also provide more comprehensive scientific evidence to inform clinical decision-making.

Based on the above, the following research hypotheses are proposed (1) Cognitive behavioral therapy has an improving effect on the negative symptoms of schizophrenia. (2) Cognitive behavioral therapy demonstrates a superior effect compared to treatment as usual (TAU) in the management of negative symptoms in schizophrenia.

## Methods

The protocol of the original review was registered in PROSPERO (number CRD: 42024579784) and published. To examine the evidence base, this meta-analysis followed the Preferred Reporting Items for Systematic Reviews and Meta-analysis (PRISMA) Statement [17]. This study is based on a meta-analysis and systematic review of published papers; hence, there is no need for an ethical statement.

### Search strategy

Literature was retrieved from 11 databases (CNKI, Wan fang Database, VIP Database, CBM Database, PubMed, EMBASE, Cochrane Library, Web of Science, APA PsycINFO, CINAHL, and MEDLINE), with the search period ranging from the inception date to 1 September 2024. The following MeSH subject headings and keywords were used: (schizophrenia OR psychosis) AND (cognitive therapy OR cognitive behavioral therapy) AND (randomized controlled trial). Reference lists of relevant reviews were also hand-searched for any further relevant studies. Please see attached for details (S1).

### Inclusion and exclusion criteria

Follow the PICOS principle in the "Cochrane Handbook for Systematic Reviews of Interventions" to determine the inclusion of studies [18]. (1) Population: Include patients aged 14 and above who have been diagnosed with schizophrenia,

 

with no gender restrictions. Exclude patients with a history of alcohol or drug dependence or intellectual disability. (2) Intervention: The experimental group received cognitive behavioral therapy, regardless of frequency and duration of intervention. (3) Comparison: The control group received treatment as usual (TAU), which included waiting for treatment, supportive counseling, routine care, mental health education, educational manuals, and other therapeutic interventions. (4) Outcome: Negative Symptoms of Schizophrenia (short-term treatment, medium-term treatment, long-term treatment), overall function, social skills, and social functioning. (5) Study design: Inclusion criteria: Only randomized controlled trials were included. Only studies that used the Positive and Negative Syndrome Scale (PANSS) for measurement were included. Only studies with treatment as usual (TAU) as the control group were included. Patients diagnosed with schizophrenia. Exclusion criteria: The study was a non-randomized controlled trial. Incorrect study population, data not reported for analyses.

## Data extractions

Based on the requirements of the study, two searchers independently and in a double-blind manner extracted and entered various data, including: the first author of the literature, year of publication, sample size of the experimental and control groups, gender, age of the participants, content of the intervention, intervention plan (duration, frequency, and cycle), outcome indicators, and other relevant data. We calculate the effect size by comparing the differences between the means of two independent samples. If the standard deviations (SDs) are not available, we calculate them based on the standard errors (SEs), confidence intervals (CIs), t-values, or p-values. We also attempt to obtain missing data by emailing the authors. If the authors have not reported the data in the study but have provided charts with data, we use GetData Digitizer version 2.20 software to extract the necessary data from the charts. If both unadjusted and adjusted data are presented in the paper, we use the adjusted data for our research.

## Quality assessment

According to the guidelines of evidence-based medicine research, the risk of bias assessment tool from the Cochrane systematic review [19] is adopted to evaluate the quality of included studies across seven indicators: random sequence generation, allocation concealment, blinding of participants and personnel, blinding of outcome assessment, incomplete outcome data, selective reporting, and other bias. In the statistical process, the quality assessment is categorized as follows: six or more indicators are considered to be at low risk of bias; three to four indicators are at moderate risk of bias; and fewer than three indicators are at high risk of bias.

## Statistical analyses

Review Manager (version 5.3.5; The Cochrane Collaboration, Copenhagen, Denmark) and the Stata software (version 14; Stata Corp, TX, USA) were used for all analyses. Researchers combine data by summing the mean values and calculating the standard deviations to derive the results after intervention. For continuous data, we calculate the mean difference (MD) and standardized mean difference (SMD) along with their corresponding 95% confidence intervals (95% CI), depending on whether the outcomes are measured by the same tool. Researchers will conduct statistical tests for heterogeneity (chi-square and I-square), with the I-square statistic used to assess the degree of heterogeneity. If $I^2 \leq 50\%$ and $P > 0.1$, a fixed-effect model will be used to pool the data; conversely, if $I^2 > 50\%$ and $P < 0.1$, a random-effects model will be employed, along with sensitivity analyses and subgroup analyses to explore the sources of heterogeneity. We have defined the subgroups for short-term treatment (closest to 3 months), medium-term treatment (closest to 6 months), and long-term treatment (closest to 1 year). We have assessed the therapeutic effects of Review Manager (version 5.3.5; The Cochrane Collaboration, Copenhagen, Denmark) and the Stata software (version 14; Stata Corp, TX, USA) were used for all analyses. Researchers combine data by summing the mean values and calculating the standard deviations to derive the results after intervention. For continuous data, we calculate the mean difference (MD) and standardized mean

difference (SMD) along with their corresponding 95% confidence intervals (95% CI), depending on whether the outcomes are measured by the same tool. Researchers will conduct statistical tests for heterogeneity (chi-square and I-square), with the I-square statistic used to assess the degree of heterogeneity. If I² ≤ 50% and P > 0.1, a fixed-effect model will be used to pool the data; conversely, if I² > 50% and P < 0.1, a random-effects model will be employed, along with sensitivity analyses and subgroup analyses to explore the sources of heterogeneity. We have defined the subgroups for short-term treatment (closest to 3 months), medium-term treatment (closest to 6 months), and long-term treatment (closest to 1 year). We have assessed the therapeutic effects of cognitive behavioral therapy on the negative symptoms of schizophrenia from the perspectives of short-term, medium-term, and long-term treatments. Finally, publication bias will be assessed using the Egger test [20], and a funnel plot will be constructed using Stata 14.0 software.

On the negative symptoms of schizophrenia from the perspectives of short-term, medium-term, and long-term treatments. Finally, publication bias will be assessed using the Egger test [20], and a funnel plot will be constructed using Stata 14.0 software.

## Quality of evidence

The certainty of the evidence is assessed by two researchers based on the five aspects of the Grades of Recommendation Assessment, Development, and Evaluation method (GRADE) approach: risk of bias, consistency, indirectness, imprecision, and publication bias. The certainty of the evidence is categorized into four levels (high, moderate, low, and very low) [21].

## Results

### Description of included studies

We conducted a thorough search of the database and identified 14504 potentially eligible studies. After reviewing for duplicates, 8794 articles were left, and then 8756 articles were excluded by reading the title and abstract carefully. Following the inclusion and exclusion criteria, 15 articles with 1311 participants were selected for this systematic review and meta-analysis by reading the full text (Fig 1).

### Characteristics of the included studies

The analysis included a total of 15 studies involving 1311 participants. The intervention was cognitive-behavioral therapy (CBT), and the sample population was predominantly male. The primary assessment scale used was the Positive and Negative Syndrome Scale (PANSS). Table 1 lists the basic characteristics of the included studies.

### Risk of bias in the included studies

The quality of the included literature was assessed, with 9 articles classified as having a low risk of bias, 4 articles scoring 7 points, and the remaining 6 articles all classified as having a moderate risk of bias (Fig 2). A graph of the proportion of bias risk in the included studies is presented (Fig 3). Most of the included literature did not provide a detailed description of the randomization method, which affects the stability of the study.

### Effect on PANSS-negative symptom

A total of 15 studies reported the effects of cognitive-behavioral therapy (CBT) on negative symptoms in schizophrenia. The experimental group included 661 participants, and the control group included 650 participants. The results showed that CBT significantly improved negative symptoms in schizophrenia and was superior to treatment as usual (TAU) (MD = -1.65, 95% CI = -2.10 to -1.21, p < 0.001, I² = 41%) (Fig 4). Four studies reported the effects of short-term CBT on negative symptoms in schizophrenia, and the results showed significant improvement in negative symptoms (MD = -2.71,

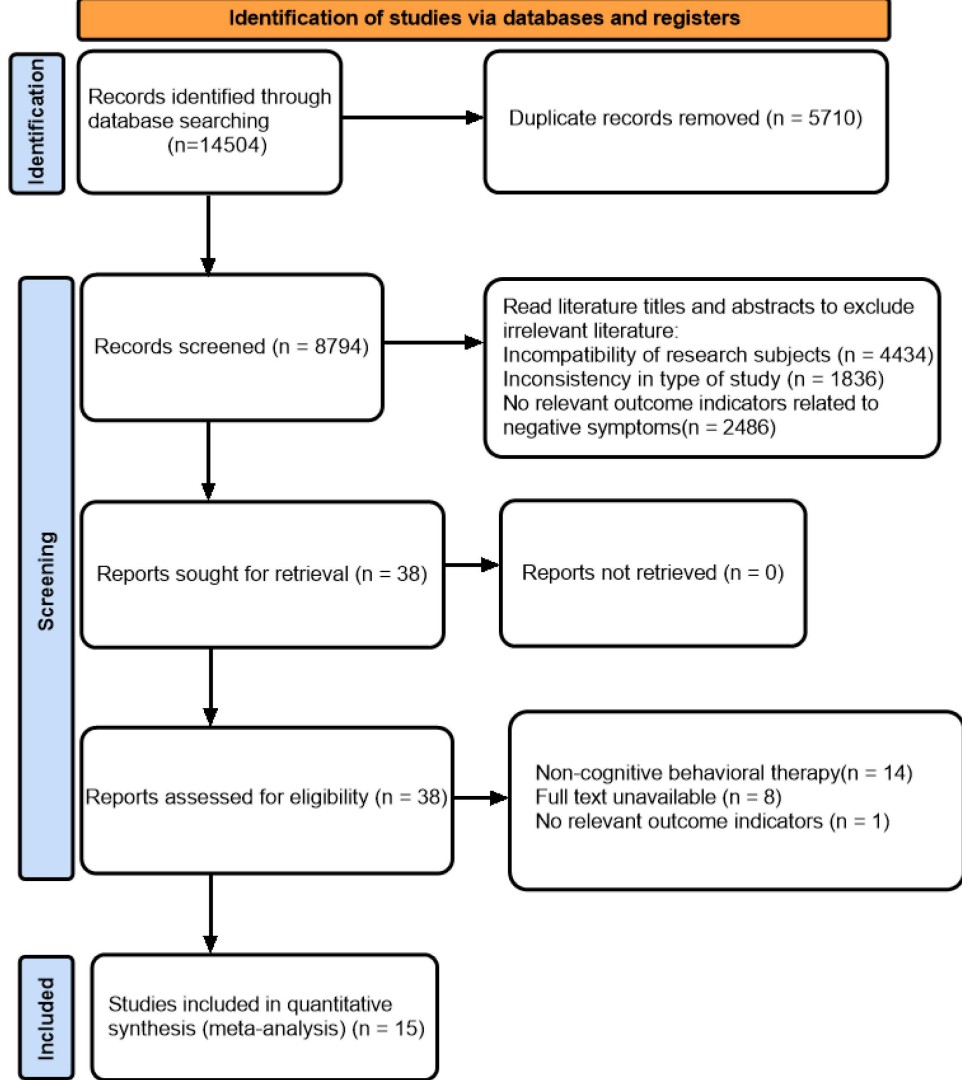

**Fig 1. PRISMA follow diagram of the study selection process.**

95% CI=-3.18 to -1.61, p<0.001, I²=48%) (Fig 5). Seven studies reported the effects of medium-term CBT on negative symptoms in schizophrenia, and the results showed significant improvement in negative symptoms (MD=-1.80, 95% CI=-2.76 to -0.84, p<0.001, I²=29%) (Fig 5). Eight studies reported the effects of long-term CBT on negative symptoms in schizophrenia, and the results showed significant improvement in negative symptoms (MD=-1.70, 95% CI=-2.54 to -0.85, p<0.001, I²=0%) (Fig 5). According to the GRADE assessment of evidence quality, the certainty of evidence for negative symptoms was rated as moderate quality, downgraded due to risk of bias (S2).

### Effect on overall function

Five research studies have reported the impact of cognitive-behavioral therapy (CBT) on overall functioning in schizophrenia. The results show that CBT can significantly improve overall functioning in patients with schizophrenia and is superior to treatment as usual (TAU) (SMD=0.38, 95% CI=0.13 to 0.63, p<0.05, I²=0%) (Fig 6). According to the GRADE

**Table 1. Characteristics of the included studies.**

| Study | Country/area | Design | Sample size (IG/CG) | Male proportion (IG: CG) | Mean age (IG: CG) | Experimental group | Control group | Duration/assessment time | Outcome measures |
|---|---|---|---|---|---|---|---|---|---|
| Barrowclough 2001 [22] | UK | RCT | 32 (17/15) | total:92% (IG:NR,CG:NR) | total:31.1(9.69) (IG:NR,CG:NR) | CBT | TAU | 18 weeks | PANSS, GAF, SFS |
| Barrowclough 2006 [23] | UK | RCT | 113 (57/56) | total:72.6% (IG:NR,CG:NR) | total:38.83(8.6) (IG:NR,CG:NR) | CBT | TAU | Total 6 months, 18 lessons, 2 hours each | PANSS, GAF, SFS |
| Morrison 2014 [24] | UK | RCT | 74 (37/37) | IG:45.9 CG:59.4 | IG:32.95(13.11) CG:29.68(11.95) | CBT+TAU | TAU | 26 treatments per week for a total of 9 months | PANSS |
| Rector 2003 [25] | Canada | RCT | 42 (24/18) | IG:62 CG:28 | IG:37.5(8.3) CG:41.2(10.9) | CBT+TAU | TAU | A total of 20 weekly treatments were administered for a total of 6 months. | PANSS |
| Penn 2011 [26] | USA | RCT | 46 (23/23) | IG:60.9 CG:60.9 | IG:23.48(3.89) CG:20.96(2.14) | CBT | TAU | 36 treatments per week for a total of 12 weeks | PANSS, GAF,SSPA |
| Gumley 2003 [27] | UK | RCT | 144 (72/72) | IG:75.0 CG:70.8 | IG:35.8(9.6) CG:36.7(10.1) | CBT | TAU | Total 52 weeks, 2–3 times per week | PANSS |
| Müller 2020 [28] | German | RCT | 25 (13/12) | IG:53.8 CG:58.3 | IG:17.46(1.51) CG:17.08(1.38) | CBT+TAU | TAU | Total 9 months, 20 sessions | PANSS, GAF |
| Sönmez 2020 [29] | Norway | RCT | 63 (32/31) | IG:53.1 CG:64.5 | IG:28.6(19–51) CG:27.1(18–43) | CBT | TAU | Total 6 months, 26 treatments, 45–60 minutes per week | PANSS, GAF |
| Peters 2010 [30] | UK | RCT | 74 (36/38) | IG:72.2 CG:52.6 | IG:34(9.8) CG:39.6(10.2) | CBT | TAU | Total of 6 months, average of 16(8–28) treatments | PANSS |
| Tarrier 2004 [31] | UK | RCT | 77 (37/40) | IG:65 CG:65 | IG: 29.5 (19.7–41.3) CG: 25.9 (21.4–35.1) | CBT | TAU | A total of 3 months, with an average treatment duration of 8 hours per session | PANSS |
| Anthony 2018 [32] | UK | RCT | 487 (242/245) | IG:73 CG:71 | IG:42.2(10.7) CG:42.8(10.4) | CBT | TAU | Total 9 months, at least 1 session per week, 60 minutes each time | PANSS, PSP |
| Anthony(2) 2018 [33] | UK | RCT | 50 (26/24) | IG:62 CG:54 | IG:23.19(6.32) CG:23.21(4.97) | CBT | TAU | Total 6 months, at least 1 session per week, 60 minutes each time | PANSS |
| Jun Yan 2024 [34] | China | RCT | 100 (50/50) | IG:100 CG:100 | IG:60.42(3.26) CG:60.20(3.24) | CBT | TAU | Total 6 months, at least 2 sessions per week, 60 minutes each time | PANSS, SSFPI |
| Dandan Chen 2024 [35] | China | RCT | 60 (30/30) | IG:56.7 CG:60 | IG:35.84(6.78) CG:36.84(6.78) | CBT | TAU | Total 12 weeks, at least 3 sessions per week, 30–45 minutes each time | PANSS, SSC |
| Faxia Peng 2023 [36] | China | RCT | 80 (40/40) | IG:NR CG:NR | IG: NR CG: NR | CBT | TAU | Total 3 months, at least 1 session per week, 30–45 minutes each time | PANSS |

Abbreviations: RCT=Randomized Controlled Trial; IG=Intervention Groups; CG=Control Group; NR=Not Reported; CBT=Cognitive Behavioral Therapy; TAU=Treatment as Usual; PANSS=Positive and Negative Symptom Scale; GAF=Global Assessment of Functioning; SFS=Social Functioning Scale; SSPA=Social Skills Performance Assessment; PSP=Personal and Social Performance Scale; SSFPI=Scale of Social Function of Psychosis Inpatients; SSC=Social Skills Checklist.

assessment of evidence quality, the certainty of evidence for overall functioning is rated as low quality, downgraded due to risk of bias and imprecision (S2).

## Effect on social skills

Three research studies have reported the impact of cognitive-behavioral therapy (CBT) on the improvement of social skills in schizophrenia. The results show that CBT can significantly improve social skills in patients with schizophrenia and is superior to treatment as usual (TAU) (SMD=0.87, 95% CI=0.58 to 1.16, p<0.001, I²=0%) (Fig 7).According to the GRADE assessment of evidence quality, the certainty of evidence for overall functioning is rated as low quality, downgraded due to risk of bias and imprecision (S2).

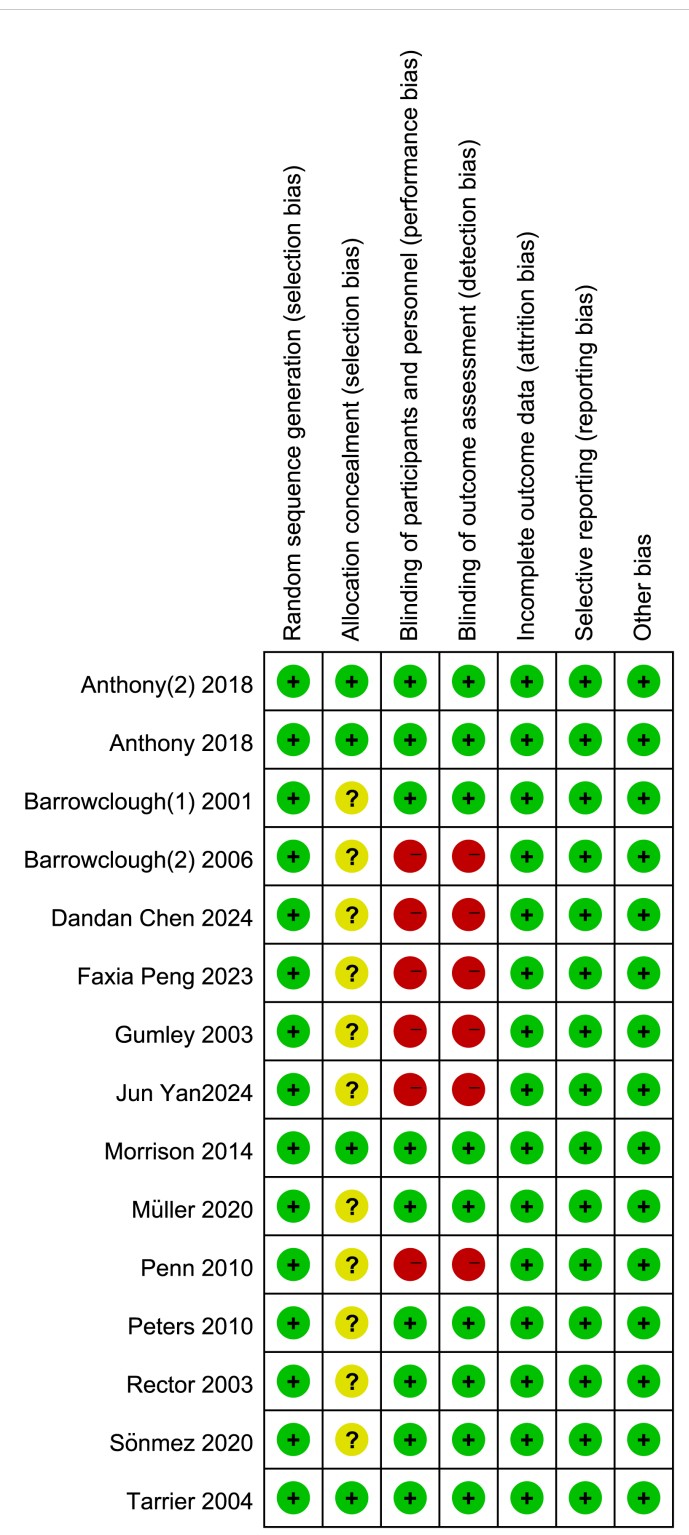

**Fig 2. Summary of risk of bias for included studies.**

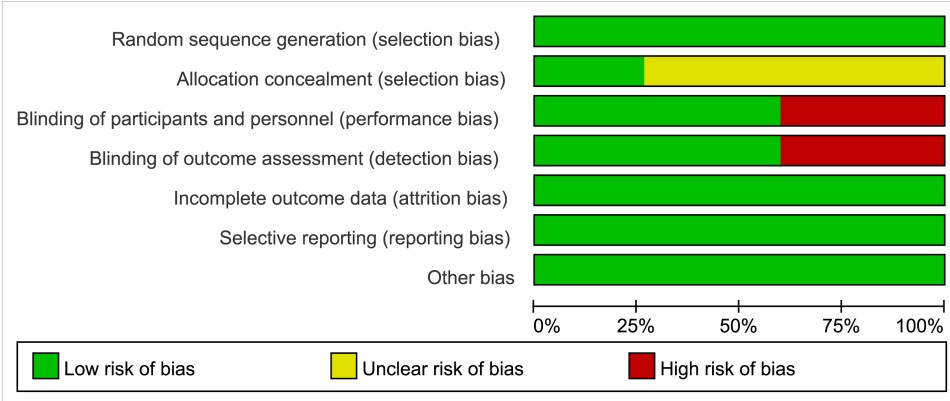

**Fig 3. Bias risk proportion graph for included studies.**

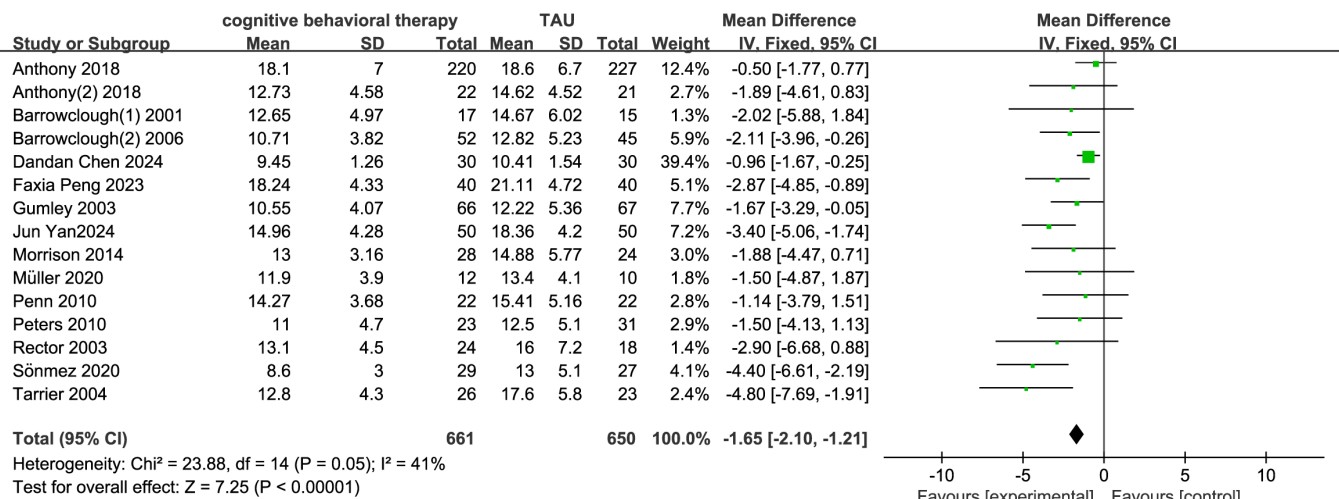

**Fig 4. Forest plot of the effect of CBT on negative symptoms of schizophrenia.**

## Effect on social functioning

Three studies have reported the impact of cognitive-behavioral therapy (CBT) on social functioning in schizophrenia. The results show that CBT can significantly improve social functioning in patients with schizophrenia and is superior to treatment as usual (TAU) (SMD = 0.19, 95% CI = 0.03 to 0.36, p < 0.05, I² = 24%) (Fig 8). According to the GRADE assessment of evidence quality, the certainty of evidence for social functioning is rated as low quality, downgraded due to risk of bias and imprecision (S2).

## Publication bias

Due to the inclusion of more than 10 studies in this research (n = 15), a publication bias test was conducted to assess the impact of cognitive therapy on improving negative symptoms [37]. The funnel plot displayed symmetry [20], indicating that the results of this study are not influenced by publication bias (S3).

|  | cognitive behavioral therapy | | | TAU | | | | Mean Difference | Mean Difference |
| Study or Subgroup | Mean | SD | Total | Mean | SD | Total | Weight | IV, Fixed, 95% CI | IV, Fixed, 95% CI |
| --- | --- | --- | --- | --- | --- | --- | --- | --- | --- |
| **1.12.1 Short-term treatment** | | | | | | | | | |
| Anthony(2) 2018 | 14 | 4.32 | 21 | 14.83 | 3.1 | 23 | 6.0% | -0.83 [-3.07, 1.41] | |
| Jun Yan2024 | 14.96 | 4.28 | 50 | 18.36 | 4.2 | 50 | 10.9% | -3.40 [-5.06, -1.74] | |
| Morrison 2014 | 13 | 3.16 | 28 | 14.88 | 5.77 | 24 | 4.5% | -1.88 [-4.47, 0.71] | |
| Tarrier 2004 | 12.8 | 4.3 | 26 | 17.6 | 5.8 | 23 | 3.6% | -4.80 [-7.69, -1.91] | |
| Subtotal (95% CI) | | | 125 | | | 120 | 25.0% | -2.71 [-3.81, -1.61] | |
| Heterogeneity: Chi² = 5.77, df = 3 (P = 0.12); I² = 48% | | | | | | | | | |
| Test for overall effect: Z = 4.84 (P < 0.00001) | | | | | | | | | |
| | | | | | | | | | |
| **1.12.2 Medium-term treatment** | | | | | | | | | |
| Anthony(2) 2018 | 14.14 | 5.47 | 22 | 14.91 | 4.72 | 22 | 3.3% | -0.77 [-3.79, 2.25] | |
| Barrowclough(2) 2006 | 13 | 4.81 | 54 | 13.31 | 5.22 | 45 | 7.6% | -0.31 [-2.30, 1.68] | |
| Morrison 2014 | 12.5 | 3.38 | 22 | 14.26 | 4.21 | 23 | 6.1% | -1.76 [-3.99, 0.47] | |
| Penn 2010 | 14.27 | 3.68 | 22 | 15.41 | 5.16 | 22 | 4.3% | -1.14 [-3.79, 1.51] | |
| Peters 2010 | 11.2 | 5.5 | 25 | 12.8 | 6 | 30 | 3.3% | -1.60 [-4.64, 1.44] | |
| Rector 2003 | 13.1 | 4.5 | 24 | 16 | 7.2 | 18 | 2.1% | -2.90 [-6.68, 0.88] | |
| Sönmez 2020 | 8.6 | 3 | 29 | 13 | 5.1 | 27 | 6.2% | -4.40 [-6.61, -2.19] | |
| Subtotal (95% CI) | | | 198 | | | 187 | 32.8% | -1.80 [-2.76, -0.84] | |
| Heterogeneity: Chi² = 8.48, df = 6 (P = 0.20); I² = 29% | | | | | | | | | |
| Test for overall effect: Z = 3.67 (P = 0.0002) | | | | | | | | | |
| | | | | | | | | | |
| **1.12.3 Long-term treatment** | | | | | | | | | |
| Anthony(2) 2018 | 12.73 | 4.58 | 22 | 14.91 | 4.72 | 22 | 4.0% | -2.18 [-4.93, 0.57] | |
| Barrowclough(1) 2001 | 12.65 | 4.97 | 17 | 14.67 | 6.02 | 15 | 2.0% | -2.02 [-5.88, 1.84] | |
| Barrowclough(2) 2006 | 10.71 | 3.82 | 52 | 12.82 | 5.23 | 45 | 8.8% | -2.11 [-3.96, -0.26] | |
| Gumley 2003 | 14.27 | 3.68 | 66 | 15.41 | 5.16 | 67 | 13.0% | -1.14 [-2.66, 0.38] | |
| Morrison 2014 | 12.61 | 4.24 | 18 | 15.95 | 5.89 | 21 | 3.0% | -3.34 [-6.53, -0.15] | |
| Müller 2020 | 11.9 | 3.9 | 12 | 13.4 | 4.1 | 10 | 2.7% | -1.50 [-4.87, 1.87] | |
| Penn 2010 | 14.27 | 3.68 | 22 | 15.41 | 5.16 | 22 | 4.3% | -1.14 [-3.79, 1.51] | |
| Peters 2010 | 11 | 4.7 | 23 | 12.5 | 5.1 | 31 | 4.4% | -1.50 [-4.13, 1.13] | |
| Subtotal (95% CI) | | | 232 | | | 233 | 42.2% | -1.70 [-2.54, -0.85] | |
| Heterogeneity: Chi² = 2.08, df = 7 (P = 0.96); I² = 0% | | | | | | | | | |
| Test for overall effect: Z = 3.94 (P < 0.0001) | | | | | | | | | |
| | | | | | | | | | |
| Total (95% CI) | | | 555 | | | 540 | 100.0% | -1.98 [-2.53, -1.44] | |
| Heterogeneity: Chi² = 18.60, df = 18 (P = 0.42); I² = 3% | | | | | | | | | |
| Test for overall effect: Z = 7.08 (P < 0.00001) | | | | | | | | | |
| Test for subgroup differences: Chi² = 2.27, df = 2 (P = 0.32), I² = 12.1% | | | | | | | | | |

-4 -2 0 2 4
Favours [experimental] Favours [control]

**Fig 5. Forest plot of the short- to medium- to long-term effects of CBT on negative symptoms in schizophrenia.**

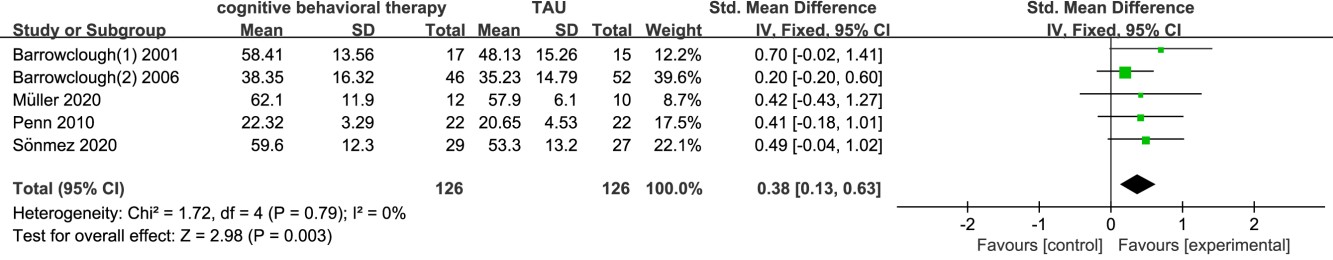

|  | cognitive behavioral therapy | | | TAU | | | | Std. Mean Difference | Std. Mean Difference |
| Study or Subgroup | Mean | SD | Total | Mean | SD | Total | Weight | IV, Fixed, 95% CI | IV, Fixed, 95% CI |
| --- | --- | --- | --- | --- | --- | --- | --- | --- | --- |
| Barrowclough(1) 2001 | 58.41 | 13.56 | 17 | 48.13 | 15.26 | 15 | 12.2% | 0.70 [-0.02, 1.41] | |
| Barrowclough(2) 2006 | 38.35 | 16.32 | 46 | 35.23 | 14.79 | 52 | 39.6% | 0.20 [-0.20, 0.60] | |
| Müller 2020 | 62.1 | 11.9 | 12 | 57.9 | 6.1 | 10 | 8.7% | 0.42 [-0.43, 1.27] | |
| Penn 2010 | 22.32 | 3.29 | 22 | 20.65 | 4.53 | 22 | 17.5% | 0.41 [-0.18, 1.01] | |
| Sönmez 2020 | 59.6 | 12.3 | 29 | 53.3 | 13.2 | 27 | 22.1% | 0.49 [-0.04, 1.02] | |
| | | | | | | | | | |
| Total (95% CI) | | | 126 | | | 126 | 100.0% | 0.38 [0.13, 0.63] | |
| Heterogeneity: Chi² = 1.72, df = 4 (P = 0.79); I² = 0% | | | | | | | | | |
| Test for overall effect: Z = 2.98 (P = 0.003) | | | | | | | | | |

-2 -1 0 1 2
Favours [control] Favours [experimental]

**Fig 6. Forest plot of the effect of CBT on overall functioning in schizophrenia.**

## Discussion

Schizophrenia is a complex and chronic mental disorder characterized by symptoms that can be broadly categorized into positive and negative symptoms [38]. Unlike the more easily identifiable positive symptoms such as hallucinations and delusions, negative symptoms manifest as blunted affect, alogia, avolition, asociality, and anhedonia [3,4]. These

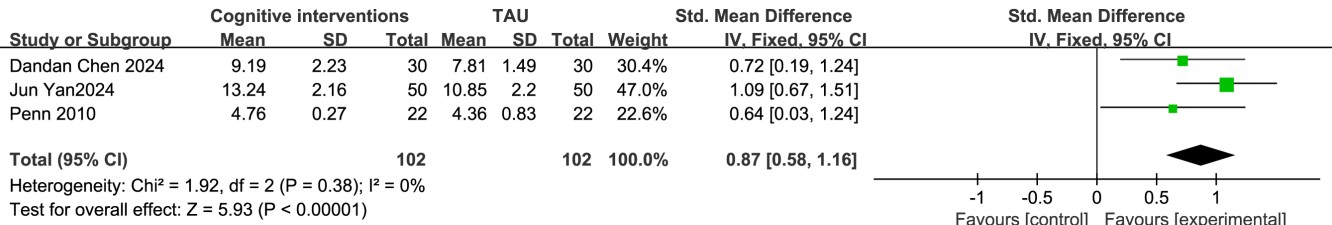

**Fig 7. Forest plot of the effects of CBT on social skills in schizophrenia.**

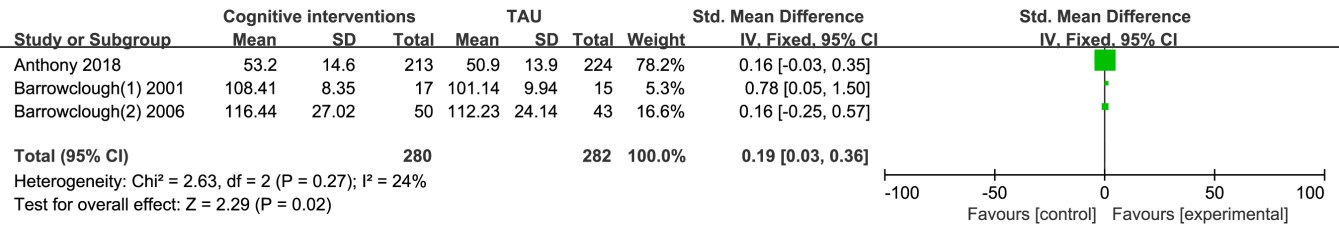

**Fig 8. Forest plot of the effects of CBT on social functioning in schizophrenia.**

symptoms are often more insidious but have a profound impact on the patient's social functioning and quality of life, and can be even more challenging to treat than positive symptoms [39]. However, current evidence suggests that antipsychotic medications have limited efficacy in reducing the severity of negative symptoms, and there is a lack of robust evidence supporting their effectiveness in treating primary and persistent negative symptoms [7–9]. This represents a core challenge in the current research on the treatment of negative symptoms and highlights the insufficient evidence supporting existing interventions. With the evolving trends in contemporary healthcare, cognitive behavioral therapy has gained increasing attention due to its safety, efficacy, minimal side effects, and high patient acceptance, and is being increasingly applied in the clinical treatment of various mental disorders [40]. Therefore, in this meta-analysis, we provide a comprehensive review of the application and efficacy of cognitive behavioral therapy in the treatment of negative symptoms in schizophrenia.

The findings of this systematic review suggest that CBT is associated with reductions in total PANSS scores, which includes improvements across positive, negative, and general psychopathology domains. Notably, the observed improvement in negative symptoms raises an important question: whether these changes reflect a direct effect on primary negative symptoms or are secondary to improvements in other symptom domains, such as positive symptoms or general functioning. This distinction is critical for understanding the mechanisms underlying the therapeutic effects of CBT in schizophrenia.

The reduction in negative symptoms observed in this review may, at least in part, be attributed to improvements in secondary negative symptoms. Secondary negative symptoms are often conceptualized as a consequence of other factors, such as the burden of positive symptoms, medication side effects, or social disengagement due to impaired functioning [3]. For instance, alleviation of positive symptoms through CBT may reduce emotional withdrawal or avolition by diminishing the distress and preoccupation associated with hallucinations or delusions [41,42]. However, it is also possible that CBT exerts a direct effect on primary negative symptoms. Primary negative symptoms are thought to arise from the core pathophysiology of schizophrenia, including deficits in neural circuits involved in motivation, reward processing, and emotional expression [43]. CBT interventions that focus on enhancing goal-directed behavior, improving emotional regulation, and challenging defeatist beliefs may directly target motivational and emotional impairments characteristic of primary

negative symptoms [44]. For example, by addressing cognitive distortions and promoting adaptive coping strategies, CBT may enhance neural substrates underlying motivation and emotional processing.

This interpretation aligns with previous research suggesting that improvements in negative symptoms are often mediated by changes in other domains. Future research should aim to disentangle the direct and indirect effects of CBT on primary and secondary negative symptoms, using more specific assessment tools and study designs.

The current review cannot definitively disentangle the relative contributions of direct and indirect effects on negative symptoms. However, the observed reductions in total PANSS scores suggest that improvements in negative symptoms are likely intertwined with broader clinical and functional gains. Future studies should employ more nuanced assessments of negative symptoms, such as the use of scales that differentiate between primary and secondary negative symptoms (e.g., the Clinical Assessment Interview for Negative Symptoms [CAINS] or the Brief Negative Symptom Scale [BNSS]). Additionally, mechanistic studies using neuroimaging or biomarkers could help clarify whether CBT directly modulates the neural circuits implicated in primary negative symptoms or whether its effects are mediated by changes in other symptom domains.

In conclusion, while the findings of this review support the efficacy of CBT in reducing negative symptoms, the extent to which these improvements reflect direct effects on primary negative symptoms versus secondary effects remains unclear. This distinction has important implications for optimizing therapeutic interventions and tailoring treatments to the specific needs of individuals with schizophrenia. Future research should aim to elucidate the mechanisms underlying these improvements to inform the development of more targeted and effective interventions.

The findings of this review highlight the potential of CBT in addressing negative symptoms in schizophrenia. However, the complexity and heterogeneity of these symptoms underscore the need for further research and innovation in both assessment and treatment. Future studies should prioritize several key areas to advance our understanding and management of negative symptoms.

First, there is a critical need for more precise and nuanced assessment tools that can differentiate between primary and secondary negative symptoms. Current scales, such as the Positive and Negative Syndrome Scale (PANSS), often conflate these dimensions, limiting our ability to disentangle their underlying mechanisms [3]. The development and validation of specialized instruments, such as the Clinical Assessment Interview for Negative Symptoms (CAINS) and the Brief Negative Symptom Scale (BNSS), represent important steps forward [45,46]. These tools should be widely adopted in clinical trials to ensure that interventions are accurately targeting the intended symptom domains.

Second, future research should focus on developing and implementing personalized intervention strategies tailored to the individual needs of patients with schizophrenia. Given the heterogeneity of negative symptoms and their varying impact on functioning, a "one-size-fits-all" approach is unlikely to be effective [47]. Personalized interventions could involve the use of predictive biomarkers, clinical profiles, and patient preferences to design treatment plans that optimize outcomes. For example, patients with prominent motivational deficits might benefit more from behavioral activation therapies, while those with severe social withdrawal might respond better to social skills training or group-based interventions [48]. Advances in digital health technologies, such as mobile apps and wearable devices, could further support personalized care by providing real-time monitoring and adaptive interventions based on individual progress [49].

Third, there is a need to expand the range of targeted interventions for negative symptoms. Although CBT has shown promise in addressing these symptoms, additional approaches should be explored to address their multifaceted nature. For instance, novel psychosocial interventions, such as acceptance and commitment therapy (ACT) or mindfulness-based therapies, could help patients manage emotional blunting and improve psychological flexibility [50]. Additionally, physical exercise programs and nutritional interventions have shown potential in improving motivation and overall well-being in individuals with schizophrenia [51,52]. Combining these approaches with traditional therapies like CBT could create a more comprehensive treatment framework that addresses both the psychological and physiological aspects of negative symptoms.

Finally, future research should emphasize the importance of long-term studies to evaluate the sustained effects of interventions on negative symptoms and functional outcomes. Longitudinal designs could help identify predictors of treatment response and inform the development of maintenance strategies to prevent symptom relapse [53]. Furthermore, studies should explore the role of environmental and social factors, such as family support and community integration, in enhancing the effectiveness of interventions [54]. By addressing these factors, researchers and clinicians can develop more holistic approaches that not only reduce negative symptoms but also improve overall quality of life for individuals with schizophrenia.

In conclusion, while significant progress has been made in understanding and treating negative symptoms in schizophrenia, much work remains to be done. Future research should focus on refining assessment tools, developing personalized interventions, expanding the range of targeted therapies, and evaluating long-term outcomes. By leveraging advances in technology and adopting a patient-centered approach, we can move closer to delivering more effective and tailored interventions that address the full spectrum of this debilitating condition.

## Challenges and limitations

This meta-analysis has several limitations that should be considered when interpreting the results. First, the included studies employed a variety of CBT protocols, with differences in duration, intensity, and content. Second, many of the studies had small sample sizes, which may limit the reliability of the findings and reduce the ability to detect significant effects. Larger, well-powered studies are needed to confirm these results. Third, most studies relied on the PANSS to assess negative symptoms, which does not distinguish between primary and secondary negative symptoms. As a result, it remains unclear whether the observed improvements are due to direct effects on primary symptoms or indirect effects mediated by changes in other domains, such as positive symptoms or functional capacity. Fourth, current interventions often adopt a standardized approach, which may not adequately address the individual needs of patients. Finally, the generalizability of the findings may be limited, as many studies included specific patient groups (e.g., stable outpatients). More diverse and representative samples are needed to ensure that the results apply to a broader population of individuals with schizophrenia

## Conclusions

The results of this study indicate that Cognitive Behavioral Therapy (CBT) is significantly effective in improving negative symptoms in patients with schizophrenia and is markedly superior to Treatment as Usual (TAU). Moreover, CBT administered in the short term, medium term, and long term can all consistently and significantly improve negative symptoms. Additionally, CBT can also significantly enhance overall functioning, social skills, and social functioning in patients with schizophrenia. Future research should focus on developing and evaluating CBT specifically targeting negative symptoms, providing further reliable evidence and promoting the application of these research findings in clinical practice.

## Supporting information

**S1. Search strategy.**
(DOCX)

**S2. GRADE assessment.**
(DOCX)

**S3. Publication bias assessment: funnel plot.**
(PDF)

**S4. PRISMA 2020 checklist.**
(PDF)

**S5. Detailed data.**
(XLSX)

**S6. Data extraction information.**
(XLSX)

**S7. List of excluded studies.**
(XLSX)

**S8. Methodological quality of the trials.**
(DOCX)

## Author contributions

**Data curation:** Yu Hong.

**Formal analysis:** Yu Hong.

**Funding acquisition:** Yu Hong, YiYun Chen.

**Investigation:** Yu Hong, WenFei Tan.

**Methodology:** Yu Hong.

**Project administration:** Yu Hong, YiYun Chen.

**Resources:** Yu Hong.

**Software:** Yu Hong.

**Supervision:** Yu Hong, YiYun Chen.

**Validation:** Yu Hong, YingLin Bai.

**Visualization:** Yu Hong.

**Writing – original draft:** Yu Hong.

**Writing – review & editing:** Yu Hong.

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
