## [Decision Letter · Decision Letter 0]

22 Feb 2025

PONE-D-24-37272Cognitive Therapy for the Improvement of Negative Symptoms in Schizophrenia: A Systematic Review and Meta-Analysis of Randomized Controlled TrialsPLOS ONE

Dear Dr. Hong,

Thank you for submitting your manuscript to PLOS ONE. After careful consideration, we feel that it has merit but does not fully meet PLOS ONE’s publication criteria as it currently stands. Therefore, we invite you to submit a revised version of the manuscript that addresses the points raised during the review process.

The reviewers have made a careful analysis of the current version of the manuscript and raised pertinent points, specially regarding the methodology carried out, that would need to be addressed by the authors in order to ensure overall the soundness and comparativeness of the work submitted.

We look forward to receiving your revised manuscript.

Kind regards,

Carlos Eduardo Thomaz, Ph.D.

Academic Editor

PLOS ONE

Journal Requirements:

3. As required by our policy on Data Availability, please ensure your manuscript or supplementary information includes the following:

Additional Editor Comments :

The reviewers have found that the manuscript has important contribution to the problem addressed. However, there is a number of unclear points raised by both reviewers that would need careful attention and relevant changes by the authors on the current version of the paper submitted.

Reviewers' comments:

Reviewer's Responses to Questions

**Comments to the Author**

1. Is the manuscript technically sound, and do the data support the conclusions?

Reviewer #1: Yes

Reviewer #2: Partly

2. Has the statistical analysis been performed appropriately and rigorously? 

Reviewer #1: Yes

Reviewer #2: Yes

3. Have the authors made all data underlying the findings in their manuscript fully available?

Reviewer #1: Yes

Reviewer #2: Yes

4. Is the manuscript presented in an intelligible fashion and written in standard English?

Reviewer #1: Yes

Reviewer #2: Yes

5. Review Comments to the Author

Reviewer #1: Thank you for the opportunity to review this paper.

I will suggest changing the language in conclusion section of abstract where the line “no significant differences were observed in short term,….” Appears to indicate there was no improved from cognitive treatment and should be changed to clarify there were no differences among the 3 subsets of treatment with respect to improvement of negative symptoms.

I think it would be helpful to have a conclusion and discussion section about the future, limitations as well as what else would you have liked to know at the end of the paper.

Otherwise, it is well designed study.

Reviewer #2: This is a systematic review and meta-analysis of randomized controlled trials investigating the effect of “cognitive therapies” on the symptoms of schizophrenia. The authors report a significant reduction in symptoms across multiple domains when compared to treatment as usual. Demonstrating the therapeutic effect of psychosocial interventions in schizophrenia is important, not only because they are effective in the treatment of various symptom domains, but also because they are not associated with the side effect burden of antipsychotic medications. he evidence is limited, as the authors put it, by the lack of targeted interventions and the absence of long-term studies, which are strongly needed.

I believe the current manuscript must undergo the following major revisions in order to be accepted for publication.

Major Points:

1. Although the title specifies “negative symptoms” as the outcome under investigation, the results also include positive symptoms, depressive symptoms, and functioning. I suggest that the authors address this issue by maintaining their focus solely on negative symptoms and functioning. This is because the effect of the interventions under investigation on positive symptoms has been reported elsewhere and is not the focus of this project, unless the authors intend to explore the relationship between reductions in positive symptoms and negative symptoms. In that case, the authors should demonstrate that the reduction in negative symptoms is strongly correlated with positive symptom reduction. This point is related to subsequent comments (No. 4 and 5).

2. A recently published systematic review and meta-analysis has investigated the response to cognitive behavioral therapy (CBT) for negative symptoms in patients with schizophrenia:

o Xu F, Xu S. Cognitive-behavioral therapy for negative symptoms of schizophrenia: A systematic review and meta-analysis. Medicine (Baltimore). 2024;103(36):e39572. doi:10.1097/MD.0000000000039572

That study restricted its search to CBT and included nonrandomized trials, whereas this systematic review included both CBT and cognitive remediation therapy (CRT). CRT has been investigated in an older systematic review and meta-analysis of randomized trials:

o Cella M, Preti A, Edwards C, Dow T, Wykes T. Cognitive remediation for negative symptoms of schizophrenia: A network meta-analysis. Clin Psychol Rev. 2017;52:43-51. doi:10.1016/j.cpr.2016.11.009

Both of these systematic reviews concluded that CBT and CRT are effective in reducing the burden of negative symptoms, notwithstanding the limitations of the included studies. However, the present systematic review grouped CBT and CRT together under one category of intervention and analyzed them jointly. I suggest dedicating a portion of the introduction to justifying this methodology. Specifically, why should a systematic review combine both CBT and CRT interventions when the techniques and primary focuses of these therapies are markedly different? Sharing the label “cognitive” does not warrant this combination. As the authors know, CBT targets automatic thoughts and biases that influence emotions and behavior. It involves the use of thought records and behavioral activation, aiming to reduce distress and disability associated with psychotic symptoms. CRT, on the other hand, is a training-based intervention that aims to improve cognitive processes such as attention, memory, executive function, social cognition, or metacognition to overcome cognitive deficits.

In addition to the previous point (No. 1), I believe this issue is a major concern in the current manuscript. The authors might want to either restrict their investigation solely to CRT (since a recently published meta-analysis already covers CBT) or provide a convincing rationale for grouping both CBT and CRT in a single meta-analysis.

Minor Points:

2. In the introduction, the authors state that negative symptoms encompass “psychomotor retardation and attention deficits.” This is not accurate. The latest conceptualization of the negative domain of schizophrenia encompasses five domains to better distinguish it from psychomotor symptoms and, especially, cognitive domains. Please refer to:

o Kirkpatrick B, Fenton WS, Carpenter WT Jr, Marder SR. The NIMH-MATRICS consensus statement on negative symptoms. Schizophr Bull. 2006;32(2):214-219. doi:10.1093/schbul/sbj053

o Strauss GP, Ahmed AO, Young JW, Kirkpatrick B. Reconsidering the Latent Structure of Negative Symptoms in Schizophrenia: A Review of Evidence Supporting the 5 Consensus Domains. Schizophr Bull. 2019;45(4):725-729. doi:10.1093/schbul/sby169

3. In the introduction, the authors explain that antipsychotic treatment is not “without potential side effects.” However, this is not why they have limited use in targeting negative symptoms. The actual reason is that antipsychotic medications have very limited efficacy in reducing negative symptom severity and lack established evidence for treating primary and enduring negative symptoms. This is precisely the problem facing therapeutic research for negative symptoms, that is, the lack of evidence for available interventions. Kindly refer to:

o Fusar-Poli P, Papanastasiou E, Stahl D, et al. Treatments of Negative Symptoms in Schizophrenia: Meta-Analysis of 168 Randomized Placebo-Controlled Trials [published correction appears in Schizophr Bull. 2022 May 7;48(3):721. doi: 10.1093/schbul/sbz071.]. Schizophr Bull. 2015;41(4):892-899. doi:10.1093/schbul/sbu170

o Krause M, Zhu Y, Huhn M, et al. Antipsychotic drugs for patients with schizophrenia and predominant or prominent negative symptoms: a systematic review and meta-analysis. Eur Arch Psychiatry Clin Neurosci. 2018;268(7):625-639. doi:10.1007/s00406-018-0869-3

o Remington G, Foussias G, Fervaha G, et al. Treating Negative Symptoms in Schizophrenia: an Update. Curr Treat Options Psychiatry. 2016;3:133-150. doi:10.1007/s40501-016-0075-8

4. In the results, the authors state that “The subgroup differences (P = 0.68).” I am not sure what this means. The subgroup analysis indicates that the reduction in negative symptoms is not consistently significant across time. In the abstract and discussion, the authors argue that this lack of effect across time is “due to the lack of cognitive therapy specifically targeting negative symptoms.” This is merely a conjecture. The results, in my view, suggest that CBT and CRT do not effectively improve primary and enduring negative symptoms, which are the main targets of negative symptom treatment in schizophrenia. This needs to be mentioned as a possible interpretation of the results. After all, secondary negative symptoms in schizophrenia also respond to antipsychotic medications. Please see:

o Correll CU, Schooler NR. Negative Symptoms in Schizophrenia: A Review and Clinical Guide for Recognition, Assessment, and Treatment. Neuropsychiatr Dis Treat. 2020;16:519-534. Published 2020 Feb 21. doi:10.2147/NDT.S225643

5. Also related to the previous point is the finding in this systematic review that total PANSS scores tend to decrease with CBT or CRT, which further raises the possibility that improvement in negative symptoms is linked to improvement in other domains—that is, the effect may primarily be on secondary negative symptoms. This issue needs to be discussed in detail. Unfortunately, due to this problem, the results do not advance the field of negative symptom therapeutics, except for the finding that there is no significant difference across short- and long-term comparisons, which, in my view, indicates a lack of benefit.

6. PLOS authors have the option to publish the peer review history of their article (what does this mean? ). If published, this will include your full peer review and any attached files.

**Do you want your identity to be public for this peer review?** For information about this choice, including consent withdrawal, please see our Privacy Policy .

Reviewer #1: No

Reviewer #2: No

---

## [Author Response · Author response to Decision Letter 0]

20 Mar 2025

Dear Editors and Reviewers:

Thank you for your letter and for the reviewers' comments concerning our manuscript entitled “Cognitive Therapy for the Improvement of Negative Symptoms in Schizophrenia: A Systematic Review and Meta-Analysis of Randomized Controlled Trials” (Manuscript ID: PONE-D-24-37272). Those comments are all valuable and very helpful for revising and improving our paper, as well as the important guiding significance to our research. We have studied comments carefully and have made correction which we hope to meet with approval. Revised portion are marked in red (with track and highlighted changes) in the paper. The main corrections in the paper and the responses to the reviewer’s comments are as follows. Please don’t hesitate to contact us in case there are any problems regarding this manuscript.

Reviewer #1

To begin with, we thank the reviewer for the effort and time put into the review of the manuscript, special thanks to you for your good comments and warm work earnestly.

In terms of content the following;

Comment 1:

I will suggest changing the language in conclusion section of abstract where the line “no significant differences were observed in short term,….” Appears to indicate there was no improved from cognitive treatment and should be changed to clarify there were no differences among the 3 subsets of treatment with respect to improvement of negative symptoms.

Reply 1:

Thank you for your valuable feedback. We appreciate your suggestion to clarify the language in the conclusion section of the abstract. Based on your comment, we have revised the sentence to more accurately reflect that there were no significant differences among the three subsets of treatment (short-term, medium-term, and long-term) with respect to the improvement of negative symptoms, rather than implying a lack of improvement from cognitive therapy.

The revised text now reads:

"The results indicate that cognitive behavioral therapy has a significant effect on improving the negative symptoms of schizophrenia and is markedly superior to Treatment as Usual (TAU). Moreover, all three sub-treatment approaches (short-term, medium-term, and long-term) can sustainably and significantly improve negative symptoms of schizophrenia. Future research should focus on developing and evaluating cognitive therapies targeting negative symptoms, providing more reliable evidence and applying these research findings to clinical practice."

We believe this revision addresses your concern and improves the clarity of our findings. Thank you again for your thoughtful review.

Changes in the text: (Abstract: see marked manuscript Page 4, line 77-83)

Comment 2:

I think it would be helpful to have a conclusion and discussion section about the future, limitations as well as what else would you have liked to know at the end of the paper.

Reply 2: Thank you very much for your valuable suggestions. We have added discussions on future research directions, limitations of the study, and areas for further exploration in the discussion section. These additions aim to provide readers with a more comprehensive perspective and guide future research. We greatly appreciate your contribution to enhancing the quality of our paper!

Changes in the text: (Discussion: see marked manuscript Page 14-16 line 381-494)

Comment 3: Otherwise, it is well designed study.

Reply 3: Thank you very much for your valuable comments. We truly appreciate your recognition that “it is well designed study.” Your feedback is highly important to us, and we will continue to refine our work based on your suggestions.

Thank you again for taking the time to review our manuscript. We look forward to further opportunities for discussion and improvement.

Reviewer #2

To begin with, we thank the reviewer for the effort and time put into the review of the manuscript, special thanks to you for your good comments and warm work earnestly.

General comments:

This is a systematic review and meta-analysis of randomized controlled trials investigating the effect of “cognitive therapies” on the symptoms of schizophrenia. The authors report a significant reduction in symptoms across multiple domains when compared to treatment as usual. Demonstrating the therapeutic effect of psychosocial interventions in schizophrenia is important, not only because they are effective in the treatment of various symptom domains, but also because they are not associated with the side effect burden of antipsychotic medications. he evidence is limited, as the authors put it, by the lack of targeted interventions and the absence of long-term studies, which are strongly needed.

Reply:

Thank you for your thoughtful and constructive feedback on our systematic review and meta-analysis. We greatly appreciate your recognition of the importance of demonstrating the therapeutic effects of psychosocial interventions, such as cognitive therapies, in the treatment of schizophrenia. We agree that these interventions not only show efficacy across multiple symptom domains but also offer a valuable alternative to antipsychotic medications by avoiding their associated side effect burden.

We also acknowledge your observation regarding the limitations of the current evidence, particularly the lack of targeted interventions and the absence of long-term studies. These points are indeed critical, and we have highlighted them in the discussion section of our manuscript. We fully agree that future research should prioritize the development of targeted cognitive therapies for specific symptom domains, as well as conduct long-term studies to better understand the sustained effects of these interventions.

Thank you again for your insightful comments, which have helped us strengthen the interpretation of our findings and underscore the need for further research in this area.

Comment 1:

Although the title specifies “negative symptoms” as the outcome under investigation, the results also include positive symptoms, depressive symptoms, and functioning. I suggest that the authors address this issue by maintaining their focus solely on negative symptoms and functioning. This is because the effect of the interventions under investigation on positive symptoms has been reported elsewhere and is not the focus of this project, unless the authors intend to explore the relationship between reductions in positive symptoms and negative symptoms. In that case, the authors should demonstrate that the reduction in negative symptoms is strongly correlated with positive symptom reduction. This point is related to subsequent comments (No. 4 and 5).

Reply1:

Thank you for your careful review and valuable feedback on our manuscript. We fully understand your suggestions regarding the title and the focus of our research. Indeed, while our study initially aimed to investigate the effects of cognitive therapy on negative symptoms, we also included data on positive symptoms, depressive symptoms, and functional improvement during the analysis. We agree with your perspective that the impact on positive symptoms has been extensively reported in other studies and is not the primary focus of this project.

Following your advice, we will adjust the focus of the paper to concentrate on the outcomes related to negative symptoms of schizophrenia, ensuring better alignment with the title and research objectives. We greatly appreciate this important comment, which will help us better focus on the research theme and enhance the clarity of the manuscript. If you have any further suggestions regarding the revised content, we would be more than happy to make additional improvements.

Changes in the text: (Result: see marked manuscript Page 9-11, line 248-356)

Comment 2:

A recently published systematic review and meta-analysis has investigated the response to cognitive behavioral therapy (CBT) for negative symptoms in patients with schizophrenia:

o Xu F, Xu S. Cognitive-behavioral therapy for negative symptoms of schizophrenia: A systematic review and meta-analysis. Medicine (Baltimore). 2024;103(36):e39572. doi:10.1097/MD.0000000000039572

That study restricted its search to CBT and included nonrandomized trials, whereas this systematic review included both CBT and cognitive remediation therapy (CRT). CRT has been investigated in an older systematic review and meta-analysis of randomized trials:

o Cella M, Preti A, Edwards C, Dow T, Wykes T. Cognitive remediation for negative symptoms of schizophrenia: A network meta-analysis. Clin Psychol Rev. 2017;52:43-51. doi:10.1016/j.cpr.2016.11.009

Both of these systematic reviews concluded that CBT and CRT are effective in reducing the burden of negative symptoms, notwithstanding the limitations of the included studies. However, the present systematic review grouped CBT and CRT together under one category of intervention and analyzed them jointly. I suggest dedicating a portion of the introduction to justifying this methodology. Specifically, why should a systematic review combine both CBT and CRT interventions when the techniques and primary focuses of these therapies are markedly different? Sharing the label “cognitive” does not warrant this combination. As the authors know, CBT targets automatic thoughts and biases that influence emotions and behavior. It involves the use of thought records and behavioral activation, aiming to reduce distress and disability associated with psychotic symptoms. CRT, on the other hand, is a training-based intervention that aims to improve cognitive processes such as attention, memory, executive function, social cognition, or metacognition to overcome cognitive deficits.

In addition to the previous point (No. 1), I believe this issue is a major concern in the current manuscript. The authors might want to either restrict their investigation solely to CRT (since a recently published meta-analysis already covers CBT) or provide a convincing rationale for grouping both CBT and CRT in a single meta-analysis.

Reply2:

Thank you for your valuable comments and suggestions. In response to your concerns regarding our combination of Cognitive Behavioral Therapy (CBT) and Cognitive Remediation Therapy (CRT) in the analysis, we have carefully reconsidered our approach. We acknowledge that, although both therapies share the label “cognitive,” they have distinct techniques and primary focuses, as you correctly pointed out. CBT targets automatic thoughts and biases, using thought records and behavioral activation to reduce distress and disability associated with psychotic symptoms. In contrast, CRT is a training-based intervention aimed at improving cognitive processes such as attention, memory, executive function, social cognition, or metacognition to address cognitive deficits. Given these differences, we agree that combining these interventions in a single meta-analysis may not be appropriate. Therefore, we have revised our systematic review to focus exclusively on CBT for the treatment of negative symptoms in schizophrenia. To ensure the accuracy and relevance of our findings, we have re-conducted the database search, limiting the intervention to CBT only. Specifically, we have excluded Cognitive Remediation Therapy (CRT) and restricted our investigation to Cognitive Behavioral Therapy (CBT) alone. Additionally, we have limited our analysis to Randomized Controlled Trials (RCTs) and included only studies with a control group receiving TAU (Treatment as Usual). The revised manuscript now specifically examines the efficacy of CBT for negative symptoms in schizophrenia, providing a clearer evaluation of its impact compared to TAU.

Thank you again for your valuable feedback. We believe that these changes have enhanced the rigor and clarity of our study.

Changes in the text: (Title: Cognitive-behavioral therapy for the improvement of negative symptoms and functioning in schizophrenia: a systematic review and meta-analysis of randomized controlled trials. Result: Result: see marked manuscript Page 9-11, line 248-356.)

Comment 3:

In the introduction, the authors state that negative symptoms encompass “psychomotor retardation and attention deficits.” This is not accurate. The latest conceptualization of the negative domain of schizophrenia encompasses five domains to better distinguish it from psychomotor symptoms and, especially, cognitive domains. Please refer to:

o Kirkpatrick B, Fenton WS, Carpenter WT Jr, Marder SR. The NIMH-MATRICS consensus statement on negative symptoms. Schizophr Bull. 2006;32(2):214-219. doi:10.1093/schbul/sbj053

o Strauss GP, Ahmed AO, Young JW, Kirkpatrick B. Reconsidering the Latent Structure of Negative Symptoms in Schizophrenia: A Review of Evidence Supporting the 5 Consensus Domains. Schizophr Bull. 2019;45(4):725-729. doi:10.1093/schbul/sby169

Reply3

We thank you for your careful review and valuable comments on our manuscript and the relevant literature provided. We fully agree with your comments and have made the corresponding revisions in the revised version.In the introduction section, we initially defined "negative symptoms" as "psychomotor retardation and attention deficits," which is indeed inaccurate. Based on the studies by Kirkpatrick et al. (2006) and Strauss et al. (2019), negative symptoms should encompass five core domains: blunted affect, alogia, avolition, asociality, and anhedonia. These domains aim to better distinguish negative symptoms from psychomotor symptoms and cognitive impairments.

In the revised manuscript, we have updated the introduction to clearly outline the five core domains of negative symptoms and have cited the aforementioned studies to support this perspective. We believe this revision will enhance the accuracy and scientific rigor of our work.

Once again, we sincerely appreciate your valuable input and look forward to your further feedback.

Changes in the text: They include blunted affect, alogia, avolition, asociality, and anhedonia, which can significantly impair patients' quality of life and hinder the rehabilitation process. (Introduction: see marked manuscript Page 5, line 95-97)

Comment 4:

In the introduction, the authors explain that antipsychotic treatment is not “without potential side effects.” However, this is not why they have limited use in targeting negative symptoms. The actual reason is that antipsychotic medications have very limited efficacy in reducing negative symptom severity and lack established evidence for treating primary and enduring negative symptoms. This is precisely the problem facing therapeutic research for negative symptoms, that is, the lack of evidence for available interventions. Kindly refer to:

o Fusar-Poli P, Papanastasiou E, Stahl D, et al. Treatments of Negative Symptoms in Schizophrenia: Meta-Analysis of 168 Randomized Placebo-Controlled Trials [published correction appears in Schizophr Bull. 2022 May 7;48(3):721. doi: 10.1093/schbul/sbz071.]. Schizophr Bull. 2015;41(4):892-899. doi:10.1093/schbul/sbu170

o Krause M, Zhu Y, Huhn M, et al. Antipsychotic drugs for patients with schizophrenia and predominant or prominent negative symptoms: a systematic review and meta-analysis. Eur Arch Psychiatry Clin Neurosci. 2018;268(7):625-639. doi:10.1007/s00406-018-0869-3

o Remington G, Foussias G, Fervaha G, et al. Treating Negative Symptoms in Schizophrenia: an Update. Curr Treat Options Psychiatry. 2016;3:133-150. doi:10.1007/s40501-016-0075-8

Reply4:

Thank you for your insightful comments, the provided references, and for highlighting the need to clarify the limitations of antipsychotic medications in targeting negative symptoms. We agree that the limited efficacy of antipsychotics in reducing the severity of negative symptoms, particularly primary and enduring ones, is a critical issue. This indeed represents a significant challenge in schizophrenia treatment research. In response to your feedback, we have revised the introduction and incorporated citations to the relevant literature you provided to better reflect this point. We believe these changes provide a more accurate and comprehensive discussion of the challenges in treating negative symptoms and the limitations of current interventions. Once again, we sincerely appreciate your valuable input, which has significantly improved the quality of our manuscript.

Changes i

---

## [Decision Letter · Decision Letter 1]

22 Apr 2025

PONE-D-24-37272R1Cognitive-behavioral therapy for the improvement of negative symptoms and functioning in schizophrenia: a systematic review and meta-analysis of randomized controlled trialsPLOS ONE

Dear Dr. Hong,

Thank you for submitting your manuscript to PLOS ONE. After careful consideration, we feel that it has merit but does not fully meet PLOS ONE’s publication criteria as it currently stands. Therefore, we invite you to submit a revised version of the manuscript that addresses the points raised during the review process.

Most of the points raised by both reviewers have been properly addressed in this new version of the paper submitted. However, there are still some points pending that would require careful attention and revision in order to address all remaining relevant issues already raised in previous revisions.==============================

We look forward to receiving your revised manuscript.

Kind regards,

Carlos Eduardo Thomaz, Ph.D.

Academic Editor

PLOS ONE

Journal Requirements:

Additional Editor Comments:

Most of the points raised by both reviewers have been properly addressed in this new version of the paper submitted. However, there are still some points pending that would require careful attention and revision in order to address all remaining relevant issues already raised in previous revisions.

Reviewers' comments:

Reviewer's Responses to Questions

**Comments to the Author**

1. If the authors have adequately addressed your comments raised in a previous round of review and you feel that this manuscript is now acceptable for publication, you may indicate that here to bypass the “Comments to the Author” section, enter your conflict of interest statement in the “Confidential to Editor” section, and submit your "Accept" recommendation.

Reviewer #1: (No Response)

Reviewer #2: All comments have been addressed

2. Is the manuscript technically sound, and do the data support the conclusions?

Reviewer #1: Yes

Reviewer #2: Yes

3. Has the statistical analysis been performed appropriately and rigorously? 

Reviewer #1: Yes

Reviewer #2: Yes

4. Have the authors made all data underlying the findings in their manuscript fully available?

Reviewer #1: Yes

Reviewer #2: Yes

5. Is the manuscript presented in an intelligible fashion and written in standard English?

Reviewer #1: Yes

Reviewer #2: Yes

6. Review Comments to the Author

Reviewer #1: Thank you for the opportunity to review this interesting paper and taking into account the previous comments and making changes accordingly.

Reviewer #2: I would like to thank the authors for their major revision of the manuscript and for considering my suggestion to restrict their systematic review and meta-analysis to CBT and focus on negative symptoms. This has significantly strengthened the manuscript and the conclusions drawn from their analysis. The authors have addressed all the issues raised in the previous round of review and now provide a more robust discussion that situates their results within the literature on negative symptom treatment in schizophrenia. I have a few comments I believe the authors should consider:

1) The authors continue to refer to “cognitive abilities” as constituting negative symptoms. This is incorrect, as I explained in detail in my previous review. I suggest this be revised in both the abstract and the introduction.

2) In the introduction, the authors state that “there is a notable absence of systematic reviews and meta-analyses that specifically address the improvement of negative symptoms.” This is also incorrect. As I pointed out in my earlier review, there is a recent systematic review and meta-analysis that addressed this same question:

Xu, Feifei, and Sheng Xu. “Cognitive-behavioral therapy for negative symptoms of schizophrenia: A systematic review and meta-analysis.” Medicine 103, no. 36 (2024): e39572. doi:10.1097/MD.0000000000039572.

I suggest the authors explicitly justify the contribution of their study in light of this prior work. One possible approach is to highlight the aspects uniquely addressed in the current review, such as the differentiation of short-, medium-, and long-term effects, as well as the inclusion of functioning as an outcome.

3) The authors report a mean difference in the abstract, but should clarify that this pertains to the PANSS Negative Symptom subscale.

Thank you again for the opportunity to re-review this valuable manuscript.

7. PLOS authors have the option to publish the peer review history of their article (what does this mean? ). If published, this will include your full peer review and any attached files.

**Do you want your identity to be public for this peer review?** For information about this choice, including consent withdrawal, please see our Privacy Policy .

Reviewer #1: No

Reviewer #2: **Yes: ** Mohammed A. Alarabi

---

## [Author Response · Author response to Decision Letter 1]

24 Apr 2025

Dear Editors and Reviewers:

Thank you for your letter and for the reviewers' comments concerning our manuscript entitled “Cognitive-behavioral therapy for the improvement of negative symptoms and functioning in schizophrenia: a systematic review and meta-analysis of randomized controlled trials” (Manuscript ID: PONE-D-24-37272R1). Those comments are all valuable and very helpful for revising and improving our paper, as well as the important guiding significance to our research. We have studied comments carefully and have made correction which we hope to meet with approval. Revised portion are marked in red (with track and highlighted changes) in the paper. The main corrections in the paper and the responses to the reviewer’s comments are as follows. Please don’t hesitate to contact us in case there are any problems regarding this manuscript.

Reviewer #1

Comment 1:

Thank you for the opportunity to review this interesting paper and taking into account the previous comments and making changes accordingly.

Reply 1:

Thank you very much for your positive feedback and for acknowledging the changes we have made. Your constructive comments have been invaluable in enhancing the quality of our manuscript. We truly appreciate your time and effort in reviewing our work.

Reviewer #2

To begin with, we thank the reviewer for the effort and time put into the review of the manuscript, special thanks to you for your good comments and warm work earnestly.

General comments:

I would like to thank the authors for their major revision of the manuscript and for considering my suggestion to restrict their systematic review and meta-analysis to CBT and focus on negative symptoms. This has significantly strengthened the manuscript and the conclusions drawn from their analysis. The authors have addressed all the issues raised in the previous round of review and now provide a more robust discussion that situates their results within the literature on negative symptom treatment in schizophrenia. I have a few comments I believe the authors should consider:

Reply:

Thank you very much for your positive and constructive feedback on our revised manuscript. We are pleased that you found the changes we made to be significant improvements, particularly in focusing the systematic review and meta-analysis on CBT and negative symptoms. Your suggestion has indeed strengthened the manuscript and helped us draw more robust conclusions.

We have carefully considered your additional comments and have addressed them in the latest revision of our manuscript. We believe these changes further enhance the clarity and relevance of our discussion within the context of negative symptom treatment in schizophrenia.

Thank you once again for your valuable insights and for taking the time to review our work. We truly appreciate your support and guidance.

Comment 1:

The authors continue to refer to “cognitive abilities” as constituting negative symptoms. This is incorrect, as I explained in detail in my previous review. I suggest this be revised in both the abstract and the introduction.

Reply1:

Thank you for your continued attention to detail and for bringing this important point to our attention once again. We have carefully reviewed the sections of our manuscript where we previously referred to “cognitive abilities” as constituting negative symptoms and have made the necessary revisions to correct this inaccuracy.

We have updated both the abstract and the introduction to accurately reflect the distinction between cognitive abilities and negative symptoms, ensuring that our terminology is precise and consistent with the established definitions in the literature. We appreciate your guidance on this matter and believe that these changes have further improved the accuracy and clarity of our manuscript.

Thank you once again for your valuable feedback.

Changes in the text: Negative symptoms of schizophrenia are a range of deficits or losses in mental functioning associated with the disorder, including blunted affect, alogia, avolition, asociality, and anhedonia. (Abstract: see marked manuscript Page 1-2 line 23-35)

Comment 2:

In the introduction, the authors state that “there is a notable absence of systematic reviews and meta-analyses that specifically address the improvement of negative symptoms.” This is also incorrect. As I pointed out in my earlier review, there is a recent systematic review and meta-analysis that addressed this same question:

Xu, Feifei, and Sheng Xu. “Cognitive-behavioral therapy for negative symptoms of schizophrenia: A systematic review and meta-analysis.” Medicine 103, no. 36 (2024): e39572. doi:10.1097/MD.0000000000039572.

I suggest the authors explicitly justify the contribution of their study in light of this prior work. One possible approach is to highlight the aspects uniquely addressed in the current review, such as the differentiation of short-, medium-, and long-term effects, as well as the inclusion of functioning as an outcome.

Reply2:

Thank you very much for your detailed and constructive feedback. We appreciate your patience and the effort you have put into helping us improve our manuscript.

We have carefully reviewed the introduction and have revised the statement to accurately reflect the existing literature. We now explicitly acknowledge the recent systematic review and meta-analysis by Xu and Xu (2024) and have provided a clear justification for the unique contributions of our study.

In our revised introduction, we have highlighted the specific aspects that differentiate our review from the prior work, such as the detailed differentiation of short-, medium-, and long-term effects, as well as the inclusion of functioning as an outcome. We believe these aspects provide valuable additional insights and contribute to the broader understanding of cognitive-behavioral therapy for negative symptoms in schizophrenia.

We have also added a citation to the Xu and Xu (2024) study to ensure proper acknowledgment of their work. We hope these revisions address your concerns and provide a clearer context for our study.

Thank you once again for bringing this important point to our attention and for providing the reference. Your feedback has been invaluable in enhancing the accuracy and relevance of our manuscript.

Changes in the text: (Introduction: see marked manuscript Page 6, line 148-159)

Comment 3:

The authors report a mean difference in the abstract, but should clarify that this pertains to the PANSS Negative Symptom subscale.

Reply3:

Thank you for your careful review and for pointing out the need for clarification in the abstract. We have revised the abstract to explicitly state that the reported mean difference pertains to the PANSS Negative Symptom subscale. This change ensures that our findings are clearly understood and accurately represented.

We appreciate your attention to detail and your suggestions for improving the clarity of our manuscript.

Changes in the text: The analysis included a total of 15 studies involving 1,311 participants. All studies used the Positive and Negative Syndrome Scale (PANSS) as the assessment tool for measuring negative symptoms of schizophrenia. (Abstract: see marked manuscript Page 4, line 68-70)

We sincerely appreciate the efforts of both the editors and reviewers for their valuable feedback and guidance throughout the process. Your expertise has been instrumental in improving our manuscript.

Best wishes,

Yu Hong

Guangzhou Huali College

---

## [Decision Letter · Decision Letter 2]

29 Apr 2025

Cognitive-behavioral therapy for the improvement of negative symptoms and functioning in schizophrenia: a systematic review and meta-analysis of randomized controlled trials

PONE-D-24-37272R2

Dear Dr. Hong,

We’re pleased to inform you that your manuscript has been judged scientifically suitable for publication and will be formally accepted for publication once it meets all outstanding technical requirements.

Kind regards,

Carlos Eduardo Thomaz, Ph.D.

Academic Editor

PLOS ONE

Additional Editor Comments (optional):

All comments have been properly addressed. Congratulations!

Reviewers' comments:

Reviewer's Responses to Questions

**Comments to the Author**

1. If the authors have adequately addressed your comments raised in a previous round of review and you feel that this manuscript is now acceptable for publication, you may indicate that here to bypass the “Comments to the Author” section, enter your conflict of interest statement in the “Confidential to Editor” section, and submit your "Accept" recommendation.

Reviewer #2: All comments have been addressed

2. Is the manuscript technically sound, and do the data support the conclusions?

Reviewer #2: Yes

3. Has the statistical analysis been performed appropriately and rigorously? 

Reviewer #2: Yes

4. Have the authors made all data underlying the findings in their manuscript fully available?

Reviewer #2: Yes

5. Is the manuscript presented in an intelligible fashion and written in standard English?

Reviewer #2: Yes

6. Review Comments to the Author

Reviewer #2: I would like to thank the authors for their consideration of my comments and their revision of this manuscript. This work is an important addition to the literature on the value of psychosocial interventions for patients with schizophrenia.

7. PLOS authors have the option to publish the peer review history of their article (what does this mean? ). If published, this will include your full peer review and any attached files.

**Do you want your identity to be public for this peer review?** For information about this choice, including consent withdrawal, please see our Privacy Policy .

Reviewer #2: **Yes: ** Mohammed A. Alarabi

---

## [Editor Report · Acceptance letter]

PONE-D-24-37272R2

PLOS ONE

Dear Dr. Hong,

I'm pleased to inform you that your manuscript has been deemed suitable for publication in PLOS ONE. Congratulations! Your manuscript is now being handed over to our production team.

Kind regards,

on behalf of

Prof. Carlos Eduardo Thomaz

Academic Editor

PLOS ONE